# Longitudinal Relation between Comprehensive Job Resources and Three Basic Psychological Needs at Work

**DOI:** 10.3390/ijerph19106302

**Published:** 2022-05-22

**Authors:** Łukasz Baka, Michał Szulawski, Monika Prusik, Łukasz Kapica, Andrzej Najmiec

**Affiliations:** 1Social Psychology Laboratory, Central Institute for Labour Protection—National Research Institute, 00-701 Warsaw, Poland; lukap@ciop.pl (Ł.K.); annaj@ciop.pl (A.N.); 2Social Psychology Unit, The Maria Grzegorzewska University, 02-353 Warszawa, Poland; mszulawski@gmail.com; 3Faculty of Psychology, University of Warsaw, 00-927 Warszawa, Poland; m.prusik@uw.edu.pl

**Keywords:** job resources, leadership, interpersonal relations, task resources, basic psychological needs, longitudinal study

## Abstract

This study aims to understand the long-term relation between comprehensive job resources and the three basic psychological needs at work (autonomy, relatedness and competence). The study was conducted in a progressive design on a sample of 1025 Polish human service professionals. Based on a typology of job resources, the three aggregated job resources index related to the task, leadership and interpersonal relations were created and the effects of each of them on the satisfaction and frustration of the three basic psychological needs, measured after 8 months, were tested. The analysis conducted by using of structural equation modelling showed that task resources are associated with the three basic psychological needs more strongly than two other kinds of resources and that that both leadership and interpersonal resources were related to the satisfaction and frustration of all the needs to the same extent. The results are discussed in the paradigm of the Conservative of Resources and the Self Determination theories.

## 1. Introduction

In many classic models of occupational health psychology, e.g., Demands-Control-Support (DCS) [1], Conservation of Resources (COR), [2,3], Demand-Induced Strain Compensation (DISC) [4] and Job Demands-Resources (JD-R) [5], job resources are treated as desired “goods” in the work environment. It is emphasized that they help workers to cope with job demands, reduce physiological and psychological cost of stress, help achieve work goals, and stimulate personal growth, learning and development [6]. In fact, several meta-analysis studies show that job resources are the source of many positive job-related outcomes, including job satisfaction [7,8], work engagement [9,10,11], job performance [12,13] and work motivation [14,15]. However, the mechanism by which job resources lead to these positive outcomes remains unclear. In order to understand the process better, various psychological concepts have been used in previous studies. Based on these frameworks, it has been suggested that job resources may lead directly to job-related well-being (COR theory [2,3]) or indirectly through, for instance, stimulating goal accomplishment (Goal Theory [16]), enhancing employees’ self-efficacy [17] and empowerment [18], or contributing to the satisfaction of their basic psychological needs—autonomy, competence and relatedness (Self-Determination Theory [19]).

Although the relationship between job resources and the satisfaction of basic psychological needs has been empirically confirmed [20], previous studies have been limited to measuring single (not global) job resources, e.g., social support, job control or feedback, e.g., [21,22,23,24]. According to the COR theory however, resources do not exist in isolation but, instead, they multiply each other, creating the so-called positive gain spiral or “resource caravans” [3]. Therefore, it is justified to examine them in a more comprehensive way. Moreover, in some research only general ratio of the basic psychological needs has been taken into account in the previous studies [23,24], without any distinction into the three individual needs—autonomy, competence and relatedness. Hence, it is not clear which needs may be satisfied by which types of job resources. Therefore, in this article, COR and SDT theories are combined and the relationships between the three types of resources (task, interpersonal and leadership) and three basic psychological needs are tested.

### 1.1. Job Resources in the Context of the COR Theory

According to the COR theory, the overall goal of human activity is to acquire, preserve and protect valued objects, i.e., resources [2]. They are used both for survival and to obtain other desired “goods”. One of the assumptions of the COR theory, related to investment of resources, is that in order to counteract, compensate or acquire new resources, people must invest what they already have. It reduces the risk of potential losses in the future. This assumption has several important practical implications [25]. Firstly, people with a larger pool of resources are less likely to lose them and more likely to make a profit in the future. Secondly, the initial loss of resources usually entails the loss of more, creating the so-called loss spiral. On the other hand, initial success in acquiring resources is conducive to acquiring more, generating a positive gain spiral. Additionally, last but not least, those who have resources invest them offensively, thereby expanding their pool, whereas those who lack resources adopt a defensive attitude and seek to preserve what they have [25].

Obviously, the assumptions of the COR theory also apply to employees and resources available in the workplace. Gaining job resources increases the global resource pool, making it more likely that additional resources will be subsequently acquired. That is, resources tend not to exist in isolation, but instead, they multiply each other, creating the so-called *resource caravan*. For example, high organizational justice (leadership resources) will likely foster higher feelings of influence and job control (task resources), and in turn, these will translate into more intensive higher activities in seeking social support and stronger relationships with other employees (interpersonal resources). Several studies confirmed that, in the long run, such resource caravans result in positive personal outcomes such as better coping, adaptation and well-being [26,27]. Such a comprehensive approach to job resources has not been used so far to better understand and explain satisfaction and frustration of basic human needs as the key concept of the Self Determination Theory (SDT [14]).

### 1.2. Self-Determination Theory

SDT assumes that the basic psychological needs are universal, and even though individuals may differ in desire to satisfy one need more than another, they all benefit from the satisfaction of all the three needs [19]. Within SDT, basic psychological needs are stringently defined as ‘‘nutriments that must be procured by a living entity to maintain its growth, integrity and health’’ [19] (p. 326). Metaphorically speaking, just as water, minerals, and sunshine are crucial for plants to blossom, SDT scholars regard satisfaction of the basic psychological needs to be essential for humans to actualize their potentials, to flourish, and to be protected from ill health and maladaptive functioning. Within SDT, three basic psychological needs are autonomy, relatedness, and competence.

At work, autonomy satisfaction occurs when in the professional environment one experiences psychological freedom, possibility to choose, and create, whereas autonomy frustration represents a feeling of being controlled and pressured. Competence satisfaction involves feeling effective and capable at work, to bring about desired outcomes, and to manage various challenges, whereas competence frustration denotes a feeling of inadequacy and failure at the tasks one is responsible for. Relatedness satisfaction involves the sense of being warmly connected to people from one’s work environment (e.g., co-workers, clients), whereas relatedness frustration denotes feelings of loneliness, ostracism or rejection [14,19,28]. Employees who feel part of a team and feel free to express their work-related and personal troubles are more likely to have their need for relatedness fulfilled than employees who feel lonely and lack confidants at work. Even though the Self Detemination Theory assumes that the three basic psychological needs could be satisfied by various job resources, the direct relation between the COR resources and SDT basic needs were not verified and these basic concepts, which are used to explain human motivation at work were not juxtaposed [2,15].

### 1.3. Purpose of the Current Study

The current study aims to verify the long-term relation between job resources and the three basic psychological needs. Following the recommendations of the COR theory, a broad approach to job resources was used. On the basis of the typology proposed by Berthelsen et al. [29], who divided job resources into three groups (related to task, interpersonal relations and leadership), eight different job resources were taken into account in this study, that were categorized into one of the three groups, creating the three aggregate job resources indicators—task resources (included influence at work and role clarity), interpersonal resources (included colleague support, social community and horizontal trust) and leadership resources (included quality of leadership, organizational justice, reward).

Although the link between job resources and basic needs has been studied before, the focus was mainly on resources related to the content of the work, e.g., task autonomy, skill utilization and work-related feedback, developmental opportunities [21,30], not taking into account other—important for employees—types of resources, such as interpersonal relations or leadership. A limitation of earlier studies is also the fact that they were conducted in the cross-sectional research paradigm. This type of research is criticized for the fact that the measurement of various health-related variables is carried out at the same time, so their assessment may be distorted by the “current” well-being of the respondents. Moreover, on the basis of a single measurement of variables, carried out at one time point, it is difficult to determine the cause and the result of the dependence studied [31]. In longitudinal studies, on the other hand, potential causes and effects are measured in a certain time interval, which to a greater extent allows for the formulation of cause-and-effect conclusions. Moreover, all previous studies on job resources—basic psychological needs link—were conducted in countries of North America and Western Europe, but not Eastern Europe, where working conditions are described by some authors as highly demanding [32]. Therefore, we propose a comprehensive approach to job resources and testing the relationship between three groups of resources and the three basic psychological needs (in regard to satisfaction and frustration of them) in the cross-lagged paradigm and on a large sample of Polish employees.

We hypothesize that all groups of job resources will be connected positively with satisfaction and negatively with frustration of the three psychological needs. Furthermore, we expectedthat each of the resource type is particularly strongly associated with a different basic psychological need. Thus, satisfaction of the need for autonomy included having control over work (e.g.,: the form and timing of work, the number of tasks assigned, opportunities for change, choice of coworkers) and a sense of having influence over important decisions in the organization (e.g.,: participation, having decision-making authority), therefore it may be expected that employees with high job control and role clarity (task resources) will have satisfied especially strongly the need for autonomy, and this assumption has been partially supported by earlier studies [20,33]. The employees with high task resources have more freedom of action, more opportunities to choose solutions that best suit their needs, and have a greater sense of empowerment and responsibility for their decisions [20,34]. Friendly relations with co-workers, getting the support and trust from them in the workplace (interpersonal resources) should meet the need for relatedness more strongly than other ones [33]. The sense of belonging to a team, “good” atmosphere in the group and the conviction that in difficult situations you can count on the help of colleagues and trust in them are conducive to the formation of close social ties. Some studies suggest that being appreciated by the supervisor, receiving awards and being treated with respect (leadership resources), in turn, provide a form of feedback on the quality of one’s work and seem to foster mainly the need for competence. [33,35,36]. 

There is a polemical question whether, whereas conducting research on the factors satisfying basic needs, one can simultaneously examine the frustration of these needs. Needs frustration refers to “the mechanism that links negative dimensions of the social environment to indices of compromised functioning and well-being” [37] (p. 1460). For example, Costa, Ntoumanis, and Bartholomew [38] and Vansteenkiste and Ryan [39] argue that one should not measure need frustration when researching the link between need satisfaction and different positive outcomes (such as job resources), as the needs frustration is related more to negative outcomes, such as ill-being. However, recent research has shown that need frustration cannot be considered the polar opposite of need satisfaction, and thus previous research that has focused only on need satisfaction does not offer a complete view on the dark side of work [37]. Therefore, in the presented study, we took into account both the satisfaction and frustration of the three basic needs—autonomy, relatedness, and competence. We hypothesize that psychological needs frustration will be negative connected with all the three job resources. Based on COR theory [2] and SDT [19], two general and three specific hypotheses were made.

**Hypothesis** **(H1).***Job resources (included task, interpersonal relations and leadership, T1) are positivelly related to the three psychological needs satisfaction (T2)*.

**Hypothesis** **(H1a).***Task resources (T1) are more strongly associated with high autonomy need than with two other needs (T2)*.

**Hypothesis** **(H1b).***Interpersonal resources (T1) are more strongly associated with high relatedness need than with two other needs (T2)*.

**Hypothesis** **(H1c).***Leadership resources (T1) are more strongly associated with high competence need than with two other needs (T2)*.

As for the needs frustration, we leave the hypothesis on more general level.

**Hypothesis** **(H2).***Job resources (included task, interpersonal relations and leaderships, T1) are negatively related to the three psychological needs frustration (T2)*.

## 2. Materials and Methods

### 2.1. Participants

The sample study includes 1025 Polish human service professionals, belonging to three occupational sectors: education (*n* = 333), health care (*n* = 342) and customer service (*n* = 350), χ^2^(2) = 0.43, *p* = 0.807. The criterion for selecting these professional groups was the specificity of work consisting in intensive and direct contact with other people, e.g., in relation to various forms of care, assistance, education or services. The study was conducted in two waves, with eight-month interval between the measurements, at the institutions and organizations where the respondents were employed. The first wave of the study was carried out between April and June 2020, the second one—after eight months. In both waves of the study, participants completed the same full set of questionnaires. All participants were treated according to the Helsinki Declaration’s ethical guidelines and received a hard copy of the questionnaires along with a letter explaining the purpose of the study. Full confidentiality of data and anonymity were guaranteed. Participants were asked to fill out the questionnaires and seal them in envelopes, which were subsequently collected by research assistants. Out of 2000 questionnaires distributed, 1515 (76%) were completed in the first step of the study (T1) and 1025 (51% of the original pool) in the second stage (T2). Finally, 1025 subjects were included in the analysis. The analysed group consisted of 727 women (71%) and 298 men (29%), χ^2^(1) = 188.39, *p* < 0.001, between 20 and 71 years of age (*M* = 43.54, *SD* = 10.92). The gender imbalance reflects the disproportion between women and men employed in these three occupational sectors in Poland (e.g., a large predominance of women among teachers and nurses). Work experience ranged from 1 to 50 years (*M* = 19.40, *SD* = 10.75). A one-way between-subjects ANOVA test showed significant differences in the distribution of age between three occupational groups, *F*(2, 1017) = 29.03, *p* < 0.001, η^2^ = 0.05. Participants from the health care sector were older in comparison with participants from the educational sector (*p* = 0.002), and the customer service sector (*p* < 0.001). Additionally, participants from the customer service sector were older than participants from the educational sector (*p* < 0.001). However, based on the aforementioned effect size, the magnitude of these differences was small. Additionally, there were differences in seniority levels between the three occupational groups analysed, *F*(2, 999) = 9.67, *p* < 0.001, η^2^ = 0.02. On average, the seniority levels were higher in the health care group in comparison with the educational group (*p* = 0.032), as well as the customer service group (*p* < 0.001). Again, based on the effect size, the magnitude of these differences was small.

### 2.2. Measurement

*Job resources.* The variable was measured with the COPSOQ II subscales [40] in the Polish version [41]. The aggregated indexes based on factor scores of job resources related to a task, interpersonal and leadership resources were used [29]. Task resources included two subscales: influence at work (*Do you have any influence on what you do at work?*) and role clarity (*Do you know exactly which area is your responsibility?*). Interpersonal resources included three subscales: colleague support (e.g., *How often are your colleagues willing to listen to your problems at work?*), social community at work (e.g., *Is there good cooperation between colleagues at work?*) and horizontal trust (e.g., *Do the employees in general trust each other?*). Leadership resources consisted of three subscales: quality leadership (e.g., *To what extent would you say that your immediate superior is good at solving conflicts?*), organisational justice (e.g., *Is the work distributed fairly?*) and rewards (e.g., *Is your work recognised and appreciated by the management?*). Each subscale contained three or four items, with answers from one (Always or, To a very large extent) to five (Never/Hardly ever or, To a very small extent). The analyses used the results of this questionnaire from the first measurement.

*Basic psychological needs at work.* Autonomy, relatedness, and competence satisfaction and frustration were assessed by the BPNSFS-Work Domain [42] in the Polish version [43]. The scale consists of 24 items, four items for each of the six subscales (i.e., autonomy satisfaction, autonomy frustration, relatedness satisfaction, relatedness frustration, competence satisfaction and competence frustration). Respondents answered the questions concerning their feelings about their jobs during the previous four weeks (e.g., *At work, I feel capable at what I do*.) on a 7-point response scale ranging from 1 (strongly disagree) to 7 (strongly agree). The analyses used the results of this questionnaire from the second measurement.

### 2.3. Analytical Procedure

The analytical procedure consisted of several steps. First, we checked all assumptions necessary to conduct statistical procedures (potential violation of normality, and linearity), also including extensive screening of the data (unusual or outlying cases, missing data, etc.). Then, we calculated various descriptive statistics for study variables, also in relationship to socio-demographics. Next, we tested the structure of the psychological tools COPSOQ II and BPNSFS employed, applying the first and second order confirmatory factor analysis using IBM SPSS AMOS ver. 27.0. In the last two steps, we tested our main model using structural equation modelling (SEM) with IBM SPSS AMOS ver. 27.0 with the study constructs in a latent form, and separately for each type of resources, and then we compared the magnitude of slopes between the models tested.

## 3. Results

### 3.1. Descriptive Statistics

The zero-order correlational coefficients were calculated for the main study constructs (Table 1) but also between main study constructs and socio-demographics (Table A1 in the Appendix A). As it appears, most of the socio-demographic characteristics were not highly related to study constructs (low correlational coefficients if significant) (Table A1—Appendix A). As expected, most of the study variables were intercorrelated except for the relationship between social support from colleagues and needs satisfaction and needs frustration (Table 1). The data was prechecked for the potential departures from linearity, normality of distribution, and outlying cases. No significant departures were found.

### 3.2. Main Analysis

#### Confirmatory Analysis for the Psychological Scales Used in the Study

We have started analytical work with the first and second order confirmatory factor analyses for selected subscales of the COPSOQ II and BPNSFS scales separately. According to the results, both scales had a structure close to the one postulated by the authors, as indicated among other things by the values of fit indices (Table 2). After some respecifications based on error covariances for both psychological scales had been added, the fit of the models improved, Δχ^2^(4) = 214.49, *p* < 0.001 for Model—COPSOQ II, and Δχ^2^(2) = 55.40, *p* < 0.001 for Model—BPNSFS; Table 2).

When assessing the fit of the models, we adapted the following criteria of a very good fit: IFI > 0.90, RMSEA < 0.05 and SMRM < 0.08. According to more relaxed criteria, the acceptable values are: IFI > 0.85, RMSEA < 0.08 and SMRM < 0.10 [44,45,46]. Both models were accepted even though some divergences from the ideal fit were apparent especially for the BPNSFS Model. In the–BPNSFS Model, the χ^2^ to df ratio was slightly elevated and there was a discriminatory issue for some subscales: Competence frustration and Relatedness frustration (*r* = 0.91), as well as Autonomy satisfaction and Competence satisfaction (*r* = 0.88). Since all of the other indices were acceptable, all regression weights were sufficiently high and significant, we have decided to accept the results for the BPNSFS Model. There were no discrimination issues in the case of the COPSOQ Model (Pearson’s *r* coefficient did not exceed a value of 0.85 for any pair of subscales), all regression weights were significant and sufficiently high (above 0.30), and all fit indices had acceptable values even though this is not easily achieved with more complex models and larger sample sizes. After the acceptance of the structures of both psychological tools, we decided to use latent versions of our indices in the main analysis.

Main analysis was conducted using structural equation modelling (SEM). We decided to build three separate models with latent indicators for three types of resources examined (Figure 1, Figure 2 and Figure 3). All three models had a very good fit judging by beforementioned criteria (Table 2). All regression paths were significant. Since the fit of models was already acceptable, we decided not to add any respecifications based on error covariances. Based on the results all resources were significantly positively related to needs satisfaction and were negatively related to needs frustration. This fully confirms H1 and H2. In order to test more specific hypotheses (H1a–H1c) regarding the strength of the relationship for the examined constructs we decided to compare the differences between slopes using *t*-test (Table 3). Partly allied with H1a, task resources were significantly more strongly associated with autonomy than with relatedness but this was not true for competence satisfaction. Task resources showed, however, more of a predictive power (judged by R^2^ values) in case of autonomy in comparison to the remaining two needs. Interpersonal resources were not more strongly associated with relatedness than with the other two needs which disconfirms H1b. Interpersonal resources also did not present higher predictive power for relatedness than for two remaining needs. Leadership resources were not significantly more highly related to competition than to the remaining two needs which diconfirms H1c. Leadership resources also did not have higher predictive power in case of competence in comparison to autonomy and competence.

## 4. Discussion and Conclusions

The main aim of this research project was to evaluate the relation between task, interpersonal and leadership resources and satisfaction/frustration of basic psychological needs in the work context [2,14]. Although several previous studies have confirmed clearly the connections between job resources and basic psychological needs [21,22,23,24], they did not take into account aggregated forms of resources. Meanwhile, according to COR theory, resources reinforce each other, creating the so-called positive gain spiral, so it is worth investigating them in a comprehensive manner, using more aggregated indicators. In addition, different types of job resources probably meet different types of needs, and these relationships have not been analysed in detail in previous studies. Therefore, based on the typology proposed by Berthelsen et al. [29], three groups of job resources—related to task (influence at work, role clarity), interpersonal relations (colleagues support, social community at work and horizontal trust) and leadership (quality leadership, organisational justice and rewards)—were taken into account in our study and the effects of each of them on the three basic psychological needs were tested. The study was conducted in a cross-lagged design, where job resources from first measure were associated with the basic psychological needs from the second measure.

In two general hypothesis (H1 and H2) we expected that job resources will be in different ways related to satisfaction and frustration of the three psychological needs. The results of correlational analyses and SEM supported these expectations. All three kinds of resources are positively connected to satisfaction of basic psychological needs and negatively to their frustration. However, the three specific hypotheses (H1a–H1c) were confirmed only to some extent. The first clear outcome, which is only partially congruent with our assumptions, is that the task resources are associated with the three basic psychological needs stronger than two other kinds of resources. The observed regularity holds for the associations with both satisfaction and frustration of the needs. We posited that task resources would be related mainly to the need for autonomy [20,33], and this relation was indeed strong, but similar to the relation with the need for competence. It is likely that task resources are related to the need for competence because they allow people to introduce new solutions and solve problems more creatively using their best skills and knowledge, while feeling that these skills are needed [34]. Moreover, the task resources were also more strongly associated with competence and relatedness needs than interpersonal and leadership resources.

It seems that high job control and role clarity (task resources) may be considered as more basic or important kinds of resources from the perspective of psychological needs satisfaction in the work context. Although the COR theory does not imply that some job resources are more important than other [3], it seems that high job decision latitude, having control over the amount of work assigned, how and when it is done, as well as a high degree of comprehensibility demands and goals set by the organization (by an employee), contribute to an increase in the feeling of self-efficacy [47], which in turn may translate into the degree of meeting the basic psychological needs in workplace [48]. This is most clear from the perspective of the need for autonomy, employees who have control over their tasks can act with higher ownership of their actions and choose the most tailored way or skills of conducting a specific task [20]. When it comes to competence, it may be satisfied by the feeling of self-efficacy at work. Knowing one’s job and being certain of who one is in an organization is one of the basic reasons for satisfaction of the need of competence [14,49]. The resources may also help the employees to consider solutions and solve obstacles by using skills and knowledge which they are more certain of, and at the same time feel that these skills used are needed [34]. Self-efficacy in the work context, on the other hand helps in getting more work-connected relations. This may explain the supporting role of the task resources for the need of relatedness [50]. Efficient employees often have higher work-related skills, which may be of value for other people because they help them do their job, and, as a consequence, they also satisfy their need for relatedness.

Another interesting result is that both leadership and interpersonal resources were related to the satisfaction of all the needs to the same extent. Contrary to our expectations that leadership resources might be mainly associated with need of competence [35], the interpersonal relations turned out to be connected mostly to the need of relatedness. However, these findings are partially consistent with a study, that showed having friendly relations with the people from organization, getting the support from the co-workers could satisfy all psychological needs—also the needs of competence and autonomy [51]. The atmosphere of trust and support encourages the exchange of information and skills, which can indirectly build the need for competence by making the co-workers feel that they are developing and learning. On the other hand, trust and friendly relations may leave more space for an autonomous kind of cooperation and sharing of duties and responsibilities, which satisfies the need for autonomy [52]. The result has also some practical implications. One can build engagement and motivation through satisfying basic psychological needs, through building all three kind of resources or focusing on the resource which is currently missing. It might be important especially in organizations with a flattened structure, which heavily rely on teamwork, where interpersonal resources may play increasingly important role and to some extent complement the leadership resources [52].

It turned out also that leadership resources (included respect, justice and reward) are related to all psychological needs in similar way, and not more strongly to competence. These findings are partially consistent with the results of other studies that have shown that leadership resources may be responsible for satisfying all three basic needs [35]. More specifically, leaders may strengthen the need for competences by delegating tasks which focus on learning new skills and encouraging them to use employees’ talents, the need for relatedness by encouraging collaboration, promoting high team spirit and by showing the importance of work to other people, and finally increasing autonomy through empowering, by such behaviours as granting freedom and responsibility, encouraging people to voice their own opinion [36]. A recent meta-analysis on basic needs [20] showed that positive leadership behaviours such as a leader’s autonomy and relatedness support and leader–member exchange are positively related to basic needs satisfaction.

The outcomes also proved that all three types of resources are negatively related to the basic psychological need frustration, with strongest association of the need frustration with the task resources. As suggested earlier, need frustration is not isomorphic with need dissatisfaction, and the correlates that arise from these two experiences are not equivalent [37]. More specifically, need dissatisfaction follows from a passive disregard for the basic psychological needs, whereas need frustration follows from an active thwarting of the basic psychological needs, and it is logical to conclude that these two processes may yield different outcomes for the individual. In other words, need dissatisfaction may stifle psychological integration and well-being, whereas need frustration may produce psychological fragmentation and ill-being. For example, a lack of connection with co-workers may leave the employee feeling less vital and happy at work, yet ostracism and rejection may leave the employee feeling isolated and depressed. In a similar way, not having a voice in organizational decision-making is phenomenologically different from being forced to comply with a particular decision. These theoretical and empirical assumptions show that the relation between different kinds of job resources and frustration of the needs should be treated with caution as presence or lack of resources is more connected with lower needs satisfaction and dissatisfaction. Frustration of the needs would require some additional active action towards needs frustration, not only a particular type of resource missing [30]. That said one can think of situations where resources are missing as situations which foster other actions responsible for need frustration. For, instance, a conflict situation between two co-workers, which can lead to some behaviours frustrating the need for relatedness, is more likely to happen at work where interpersonal resources are missing. If the environment of these two conflicted co-workers does not provide them with interpersonal resources (such as trust among other co-workers) the conflict may be much more frustrating for the needs for relatedness. The outcomes of this study show that all three types of resources provide an environment which fosters less needs frustration [37]

### 4.1. Limitations

This research is not without limitations. Although the study was designed in the cross-lagged paradigm, we still cannot interpret the results as cause-and-effect [53]. In the future, to complement this study one might consider designing the experiment study with planned manipulation of the resources and measure of the basic psychological needs. Another limitation is that the data were gathered using questionnaires without any objectively measured data. In addition, we relied on self-reported variables measured from the same source. When measuring several variables with the self-report method, there is a risk of occurrence of the common bias method, consisting in artificially inflating the correlation coefficient between the examined variables. When considering generalisability, it should be noted that the results of this study were obtained from human service employees. The observed regularities relate to this kind of professions only and should not be generalised to other occupations and market sectors. The final issue is the gender disproportion in the research sample. Women were overrepresented because the number of women in the human service is significantly greater [54]. For the male population, in traditionally typical male occupations, the results would be perhaps different.

Another issue is that the research was conducted during the COVID-19 pandemic; hence, some responses (e.g., related to social support at work and co-worker trust) may be biased by the specificity of the current situation. During a pandemic, the organisation of work and the level of job demands are different from traditional ones—for example, a significant proportion of employees work remotely, so interpersonal relations at work are significantly limited. These unusual job conditions may have a considerable impact on the results obtained.

### 4.2. Suggestions for Future Research

Apart from some limitations, the study adds to knowledge by the detailed investigation of the links between three groups of job resources and satisfaction/frustration of the three basic psychological needs in important human service professions in Eastern Europe. In future studies, it would be worth investigating the regulation role of personal resources in job resources—basic needs links. Taken into consideration the job-related context of research, research on those specific types of personal resources that strictly relate to professional functioning of employees would be particularly useful (e.g., job-related self-efficacy, occupational hardiness or occupational resilience). As suggested by some authors, these kinds of specific job-related factors are particularly effective in the development of well-being at work [55]. In the future, it would be also advisable to measure basic psychological needs as mediating variables between job/personal resources and other positive work-related variables such as motivation, effectiveness or job crafting. Research in which job resources, psychological needs and work outcomes are measured at three time points would be particularly desirable.

## Figures and Tables

**Figure 1 ijerph-19-06302-f001:**
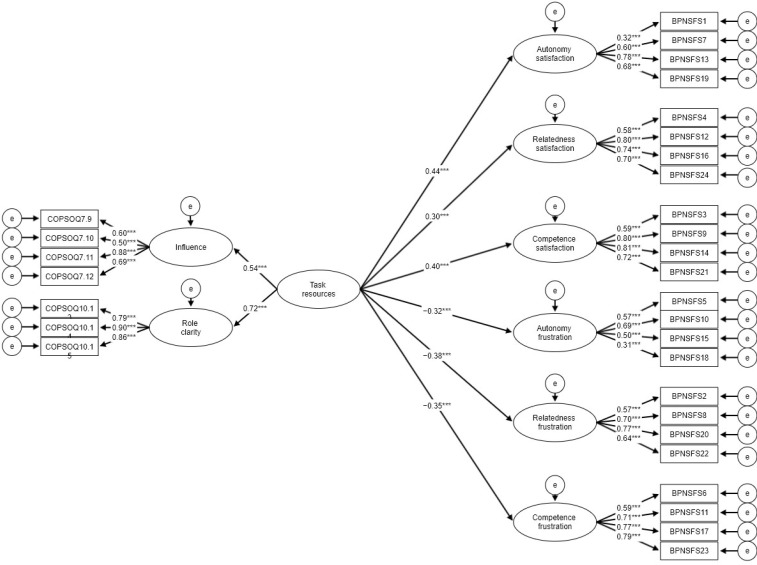
SEM model for needs satisfaction and frustration regressed on task resources. Coefficients for covariances were omitted for clarity; *** *p* < 0.001.

**Figure 2 ijerph-19-06302-f002:**
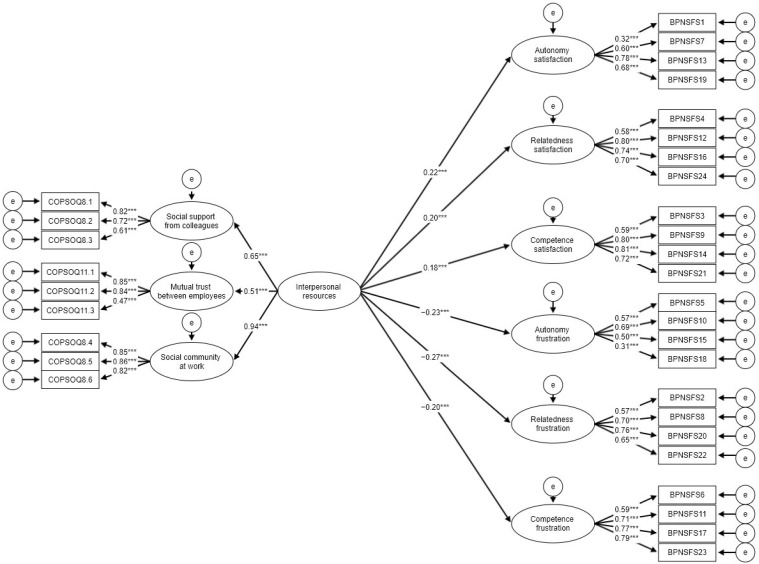
SEM model for needs satisfaction and frustration regressed on interpersonal resources. Coefficients for covariances were omitted for clarity; *** *p* < 0.001.

**Figure 3 ijerph-19-06302-f003:**
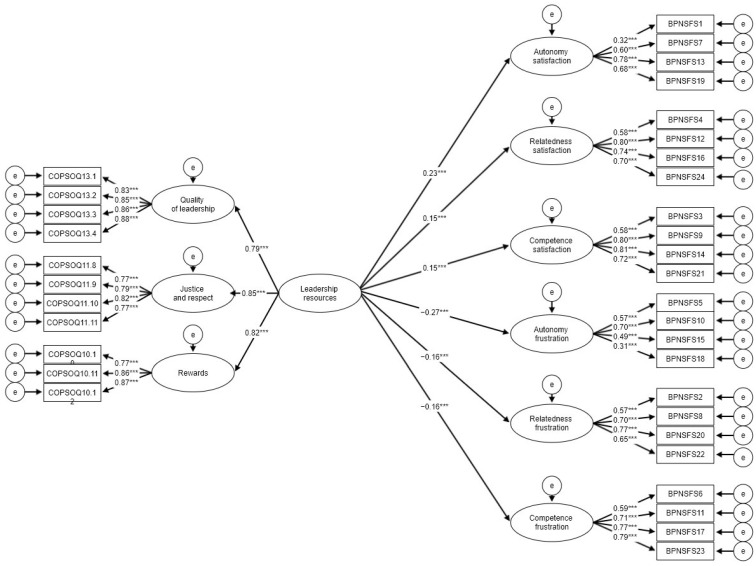
SEM model for needs satisfaction and frustration regressed on leadership resources. Coefficients for covariances were omitted for clarity; *** *p* < 0.001.

**Table 1 ijerph-19-06302-t001:** Pearson’s Correlation Coefficients for Study Variables, N = 1025.

	1	1a	1b	2	2a	2b	2c	3	3a	3b	3c	4	5	6	7	8
1. Task resources	--															
1a. Influence	0.84 ***	--														
1b. Role clarity	0.79 ***	0.33 ***	--													
2. Interpersonal resources	0.41 ***	0.22 ***	0.47 ***	--												
2a. Social support	0.17 ***	0.09 **	0.19 ***	0.78 ***	--											
2b. Mutual trust	0.32 ***	0.17 ***	0.37 ***	0.74 ***	0.33 ***	--										
2c. Social community at work	0.48 ***	0.26 ***	0.55 ***	0.83 ***	0.49 ***	0.46 ***	--									
3. Leadership resources	0.54 ***	0.35 ***	0.54 ***	0.58 ***	0.38 ***	0.49 ***	0.50 ***	--								
3a. Quality of leadership	0.34 ***	0.20 ***	0.37 ***	0.48 ***	0.38 ***	0.37 ***	0.39 ***	0.85 ***	--							
3b. Justice and respect	0.47 ***	0.34 ***	0.43 ***	0.50 ***	0.29 ***	0.48 ***	0.43 ***	0.86 ***	0.60 ***	--						
3c. Rewards	0.57 ***	0.36 ***	0.59 ***	0.51 ***	0.32 ***	0.41 ***	0.48 ***	0.86 ***	0.59 ***	0.61 ***	--					
4. Autonomy satisfaction	0.30 ***	0.24 ***	0.24 ***	0.16 ***	0.03	0.16 ***	0.19 ***	0.22 ***	0.18 ***	0.18 ***	0.21 ***	--				
5. Relatedness satisfaction	0.19 ***	0.13 ***	0.19 ***	0.13 ***	0.02	0.12 ***	0.17 ***	0.13 ***	0.09 **	0.12 ***	0.11 ***	0.62 ***	--			
6. Competence satisfaction	0.26 ***	0.17 ***	0.26 ***	0.10 ***	−0.05	0.11 ***	0.19 ***	0.13 ***	0.08 **	0.12 ***	0.12 ***	0.67 ***	0.67 ***	--		
7. Autonomy frustration	−0.17 ***	−0.11 **	−0.17 ***	−0.15 ***	−0.06 *	−0.14 ***	−0.15 ***	−0.18 ***	−0.15 ***	−0.16 ***	−0.16 ***	−0.22 ***	−0.24 ***	−0.25 ***	--	
8. Relatedness frustration	−0.21 ***	−0.09 **	−0.27 ***	−0.19 ***	−0.06	−0.14 ***	−0.25 ***	−0.14 ***	−0.10 ***	−0.13 ***	−0.12 ***	−0.34 ***	−0.52 ***	−0.52 ***	0.47 ***	--
9. Competence frustration	−0.19 ***	−0.08 *	−0.23 ***	−0.11 ***	0.01	−0.12 ***	−0.17 ***	−0.12 ***	−0.09 **	−0.10 ***	−0.11 ***	−0.38 ***	−0.48 ***	−0.64 ***	0.47 ***	0.73 ***

* *p* < 0.05, ** *p* < 0.01, *** *p* < 0.001.

**Table 2 ijerph-19-06302-t002:** Model adequacy and goodness of fit indices of the final models.

Models	*Χ* ^2^	*df*	*p*	*Χ* ^2^ */df*	RMSEA [Low, High]	SRMR	IFI	AIC	β Range [Absolute Values]
**CFA models**									
Model—COPSOQ II—Second order CFA	1654.75	313	<0.001	5.29	0.065 [0.062, 0.068]	0.08	0.92	1784.75	[0.48, 0.90]
Model—COPSOQ II—Second order CFA with modification indices	1440.26	309	<0.001	4.66	0.060 [0.057, 0.063]	0.07	0.93	1578.26	[0.45, 0.93]
Model—BPNSFS—First order CFA	1357.10	237	<0.001	5.73	0.064 [0.064, 0.071]	0.06	0.89	1483.10	[0.31, 0.81]
Model—BPNSFS—First order CFA with modification indices	1301.70	235	<0.001	5.54	0.067 [0.063, 0.070]	0.05	0.90	1431.70	[0.31, 0.82]
**Main analysis—SEM models**									
Model—Task resources	1843.12	411	<0.001	4.48	0.058 [0.058, 0.061]	0.06	0.90	2013.12	[0.30, 0.90]
Model—Interpersonal resources	1897.48	471	<0.001	4.03	0.054 [0.052, 0.057]	0.06	0.90	2077.48	[0.18, 0.94]
Model—Leadership resources	1877.63	536	<0.001	3.50	0.049 [0.047, 0.052]	0.05	0.93	2065.63	[0.15, 0.88]

Note. RMSEA = root mean square error of approximation; SRMR = standardized root mean square residual; CFI = comparative fit index; TLI = Tucker Lewis index; AIC = Akaike’s information criterion.

**Table 3 ijerph-19-06302-t003:** Comparison of slopes for models including needs satisfaction regressed on task resources, interpersonal resources, and leadership resources, N = 1025.

	Autonomy Satisfaction		Relatedness Satisfaction		Competence Satisfaction		Differences in Slopes		
							**β_1_ vs. β_2_**	**β_1_ vs. β_3_**	**β_2_ vs. β_3_**
	**β_1_**	**R^2^**	**β_2_**	**R^2^**	**β_3_**	**R^2^**	* **p** *	* **p** *	* **p** *
**Predictors**									
1. Task resources	**0.44 *****	0.19	**0.30 *****	0.09	**0.40 *****	0.16	0.565	**0.004**	**0.005**
2. Interpersonal resources	**0.22 *****	0.05	**0.20 *****	0.04	**0.18 *****	0.03	0.111	0.187	0.798
3. Leadership resources	**0.23 *****	0.05	**0.15 *****	0.02	**0.15 *****	0.02	0.969	0.551	0.647

*** *p* < 0.001; statistically significant coefficients are written in bold.

## Data Availability

The datasets generated for this study are available on request to the corresponding author.

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
