# Peer review of "Longitudinal Relation between Comprehensive Job Resources and Three Basic Psychological Needs at Work"

_ijerph, 2022, doi:10.3390/ijerph19106302_

Round 1

Reviewer 1 Report

Thank you for inviting me to review this manuscript.  It is refreshing to review a study which does not use a cross-sectional design, is theoretically grounded and the data is subject to appropriate statistical analysis.  Furthermore the results are properly interpreted and the authors appreciate the limitations of their work and share potential areas for further research.

My suggestions for improvement are simply editorial.  I have no issues with the design, analysis and interpretation of the results.

p. 3/14. Some research states (not state) that leadership resources....

p. 3/14. ......cross-sectional research paradigm

p. 4/14. .....on a large sample of Polish employees (would be better English)

p. 4/14 .......an eight month.... (word 'an' is missing)

Other comments:

In the Materials and Methods it would be helpful to make it clear which measures were taken first and which were taken 8 months later.  

In the results make it clear that Table 1A is in an Appendix.

Table 1 is very difficult to read.

Author Response

We would like to thank the Reviewer for all comments and suggestions. In line with them, we have incorporated all suggested linguistic changes. In the Materials and Methods we have made clarity which measures were taken first and which were taken after 8 months (p. 5, 6). In the results we have explained that Table 1A is in an Appendix. We also improved the clarity of all tables and figures.

Reviewer 2 Report

1. This paper does not explain a complete or clear theoretical derivation process, and I hope the author will pay attention to the writing paradigm. The author should explain in more detail what the hypothesis of this paper is. Or what is the research proposition? At the same time, I hope the author can explain in detail how the three kinds of job resources will affect the three basic psychological needs. 
2. The figure and table can't be seen clearly at all. Please pay attention to the clarity of the figure.

Author Response

We would like to thank the Reviewer for all comments. As recommended, we have refined the theoretical part of the article. First of all, we added hypotheses that were derived from psychological theories (COR and SDT). We made two general hypotheses that concern the relationships between general job resources and psychological needs satisfaction (H1) and frustration (H2); and the three specific hypotheses (H1a-H1c) concern the relations of task, interpersonal and leadership resources to satisfaction of psychological needs. Due to the fact that most of the authors emphasize that the source of frustration of needs are "bad" work conditions (not "good" job factors), we limited ourselves to the general hypothesis (H2) and we did not make specific hypotheses concern relationships between the three groups of job resources and frustration of needs. The addition of the hypotheses entailed minor changes to the Introduction and also to the Discussion. All changes were made in the track changes mode (see 3-5 and 10-12). We also added the description of results regarding Table 3. Table 3 was rearranged and some data were recalculated as we decided to test more specific hypotheses (H1a-H1c). Various additional small mistakes were corrected. We improved also the clarity of figures and tables. Some tables have still rather small font but otherwise they would not fit A4 paper size. However, we believe that these tables will be properly adjusted to the final format easily readable format by the publisher as it usually happens.

Reviewer 3 Report

Dear authors,

I am grateful for the opportunity to read this very interesting manuscript.

Please see my comments below:

Typos/ spelling:

Abstract: “…using of [double word spacing] structural equation…”

Page 5, “Analytical Procedure”: Please name software fully, “IBM SPSS AMOS 27.0” instead of “Amos ver. 27”

Tables and figures: The resolution is suboptimal

Further comments:

Please provide more information, how the study sample was drawn and constructed. Why these occupations? Why these numbers? Why did the gender imbalances occur? How could sample characteristics influence the study results?

Thank you and all the best!

Author Response

We would like to thank the Reviewer for comments. We've made all the suggested changes to the text editing, name of used software and better clarity tables and figures. We have also detailed information about the research sample. The criterion for selecting these professional groups was the specificity of work consisting in intensive and direct contact with other people, e.g. related to various forms of care, assistance, education or services. The gender imbalance reflects the disproportion between women and men employed in these three occupational sectors in Poland (e.g. a large predominance of women among teachers and nurses). As part of the work on another article, we compared the level of job resources and the degree of satisfaction and frustration psychological needs in these groups. We did not observe any differences, hence it can be assumed that the employment sector did not have a significant influence on the results obtained.

Reviewer 4 Report

The study aims to understand the relation between job resources and the three basic psychological needs, on big sample (1025) of Polish employees from 3 occupational sectors: education, health care and customer service, in regard to satisfaction and frustration of them. Propose in longitudinal study for the formulation of cause-and-effect conclusions, interesting approach and methodology used to analyze the related variables.

Results could be more explored, mainly relate to the differences of each professions  analyzed, and Figures and Table need to be expanded to better present the information.

Author Response

We would like to thank the Reviewer for all comments. As expected, we improved the quality of tables and figures. Regarding the comment on the additional analyzes of the differences between the three occupational groups, while working on another article, we compared the level of job resources and the degree of satisfaction and frustration psychological needs in these groups. We did not observe any differences, therefore we conducted statistical analyzes on the entire sample, without the division into the occupational sector. However, we added two general and three specific hypotheses to the text. The addition of the hypotheses entailed minor changes to the Introduction and also to the Discussion. All changes were made in the track changes mode (see 3-5 and 10-12). We also added the description of results regarding Table 3. Table 3 was rearranged and some data were recalculated as we decided to test more specific hypotheses (H1a-H1c).

Round 2

Reviewer 2 Report

Thank you for the opportunity to read the revised version of the manuscript.

The authors have investigated the long-term relationship between comprehensive job resources and three psychological needs at work. Drawing from related theory, they divided job resources into three groups-task resources, interpersonal resources and leadership resources. However, they did not put the three groups of resources into one SEM model. I would suggest the authors to put the three onto one SEM model to check the overall effects.

Besides, why do the authors suggest that the effect after eight months is a longitudinal effect? And based on what? Let’s say if eight months is a longitudinal effect, then the three job resources might also disappear during this period as possible changes might happen inside the organizations and among employees. So we suggest that the authors ought to measure the three resources again after eight months.

Finally, the authors integrated COR theory with the self-determination theory; however, they did not give enough explanation as to why they integrate the two theories. In my opinion, they need to make this point more clear.

Author Response

We would like to thank the Reviewer for all suggestions and comments.

(1)Indeed, one of our first analytical steps was to analyze all dependencies within one large model. We did that and the model had good fit indices, however, the model was large and difficult to interpret especially that both satisfaction and frustration was included. The most problematic thing was however, that some of the paths lost their significance (especially for interpersonal resources) which is a typical case in overly large models when different constructs are included together. In here we also had a different perspective included (satisfaction vs. frustration). At the end for the sake of clarity and transparency we decided to present everything in separate models for each type of resources. However, in the response for the Reviewer we include Amos outputs with the results for the “large” model not included in our paper.

(2) We checked that and it looks like that the levels of sources did not change at all (for interpersonal resources, t(1024) = 1.56, p = .119, d = .05; and for leadership sources, t(1024) = -1.17, p = .242, d = .04)  or they have changed (in case of task resources) in a very minimal way (very small Cohen’s d), t(1024) = 2.99, p < .05, d = .12). I hope this clarify our analytical approach.

(3) As suggested by the Reviewer, we better explained the theoretical context of the research, in particular the combination of COR and SDT theories in one study (p. 1-3).
